# Sedimentary DNA insights into Holocene Adélie penguin (*Pygoscelis adeliae*) populations and ecology in the Ross Sea, Antarctica

**Jamie R. Wood** [1,2,13] ✉, **Chengran Zhou** [3,4,13], **Theresa L. Cole** [5],
**Morgan Coleman**[5], **Dean P. Anderson** [5], **Phil O'B. Lyver**[5], **Shangjin Tan**[3,6],
**Xueyan Xiang**[3,4], **Xinrui Long**[3,7], **Senyu Luo**[3,6], **Miao Lou**[8], **John R. Southon**[9],
**Qiye Li** [3,4] & **Guojie Zhang** [10,11,12] ✉

We report 156 sediment metagenomes from Adélie penguin (*Pygoscelis adeliae*) colonies dating back 6000 years along the Ross Sea coast, Antarctica, and identify marine and terrestrial eukaryotes, including locally occurring bird and seal species. The data reveal spatiotemporal patterns of Adélie penguin diet, including spatial patterns in consumption of cnidarians, a historically overlooked component of Adélie penguin diets. Relative proportions of Adélie penguin mitochondrial lineages detected at each colony are comparable to those previously reported from bones. Elevated levels of Adélie penguin mitochondrial nucleotide diversity in upper stratigraphic samples of several active colonies are consistent with recent population growth. Moreover, the highest levels of Adélie penguin mitochondrial nucleotide diversity recovered from surface sediment layers are from the two largest colonies, indicating that *seda*DNA could provide estimates for the former size of abandoned colonies. *Seda*DNA also reveals prior occupation of the Cape Hallett Adélie penguin colony site by southern elephant seal (*Mirounga leonina*), demonstrating how terrestrial *seda*DNA can detect faunal turnover events in Antarctica driven by past climate or sea ice conditions. Low rates of cytosine deamination indicate exceptional *seda*DNA preservation within the region, suggesting there is high potential for recovering much older *seda*DNA records from local Pleistocene terrestrial sediments.

With climate and environmental change posing an imminent threat to Antarctic biota[1,2], new approaches to monitoring biodiversity and understanding the long-term dynamics of Antarctic ecosystems and their responses to past climate and environmental change events are urgently required[3,4]. The study of sedimentary ancient DNA (*seda*DNA) can provide such insights[5] by using preserved genetic signatures of past plant, animal, and microbial diversity that enable the reconstruction of spatiotemporal patterns of species distributions at population-level resolution[6–8]. Importantly, this high-resolution data can aid in the development of process-based models that permit forecasts of ecosystem responses to future climate scenarios[9]. Studies have demonstrated the utility of *seda*DNA in understanding the long-term dynamics of aquatic (marine and lacustrine) ecosystems in Antarctica, specifically by detecting biological responses of productivity

to past climate and environmental change events[10–20]. However, compared with permafrost soils and sediments from the northern hemisphere boreal regions, which have been a major focus of ancient DNA research since the mid-1990s[21–23], *seda*DNA from analogous terrestrial substrates at southern high latitudes have received relatively little attention[24]. This is despite the extensive application of environmental DNA (eDNA) tools to the characterisation and monitoring of modern Antarctic biota in terrestrial environments[25].

In Antarctica, terrestrial ice-free Late Quaternary sediments are exposed mainly in near-coastal sites. Most sediment exposures are of Holocene age, though some contain faunal remains dating back at least ca. 45,000 years[26]. Detrital material and aerosols transported onshore by winds[11] mean that *seda*DNA within these near-coastal deposits have the potential to record both terrestrial and coastal marine biodiversity. Moreover, ice-free near-coastal sites are favoured by breeding pygoscelid penguins (*Pygoscelis* spp.), whose guano, rich in marine nutrients, concentrates terrestrial biodiversity[27] and also contains remains of marine prey species[28–30]. Pygoscelid penguin colonies occur around the coastal margin of the Antarctic continent, and active colonies are typically interspersed with abandoned colonies, which were active during earlier periods of the Holocene when climatic or sea ice conditions may have been different from the present. For these reasons, previous studies of biological change in coastal Antarctica during the Holocene have focussed on stratigraphic records from pygoscelid penguin colonies, using a range of proxies including bones and otoliths[30], dietary stable isotopes[31] and other molecular markers[32]. *Seda*DNA from pygoscelid penguin colonies, therefore, offers significant potential to contribute towards understanding the recent history of penguin populations and their ecology, but also past terrestrial and marine biodiversity in Antarctica, the interaction

between these ecosystems, and their responses to climatic and environmental change during the Late Quaternary.

Here, we sequenced metagenomes of 156 stratigraphically-sampled sediment samples collected from fourteen pits excavated at six active and four abandoned Adélie penguin (*Pygoscelis adeliae*) colonies (Fig. 1A; Supplementary Table 1) along the Ross Island and East Victoria Land coastlines, Antarctica. We analyse these data to assess Holocene eukaryotic biodiversity patterns within the Ross Sea region, focussing on Adélie penguin population history and spatio-temporal patterns of the Adélie penguin diet, and investigate the potential of terrestrial *seda*DNA archives as a resource for reconstructing past biodiversity patterns at high southern latitudes.

## Results and discussion
### Stratigraphies and chronologies

Excavated pits ranged in depth from 150 to 770 mm. Stratigraphic sequences varied between pits (Supplementary Figs. 1 and 2), but typically consisted of characteristic yellow-brown ornithogenic soil horizons overlying grey-black sands and gravels. The ornithogenic soils contained common Adélie penguin eggshell fragments, bones, feathers and desiccated chick remains. In abandoned colonies, the uppermost sediment layers consisted of grey-black gravel and sand that obscured the ornithogenic soil horizons beneath (Supplementary Figs. 1 and 2).

Colonies were grouped into four age classes based on radiocarbon dating of eggshell fragments isolated from the sediments (Supplementary Tables 1 and 2). Calibrated median ages from active colonies (Fig. 1) ranged from modern to 225 yrs BP (Supplementary Table 2). Cape Barne represents a recently abandoned colony site, with our dating indicating colony abandonment within the last 400 years

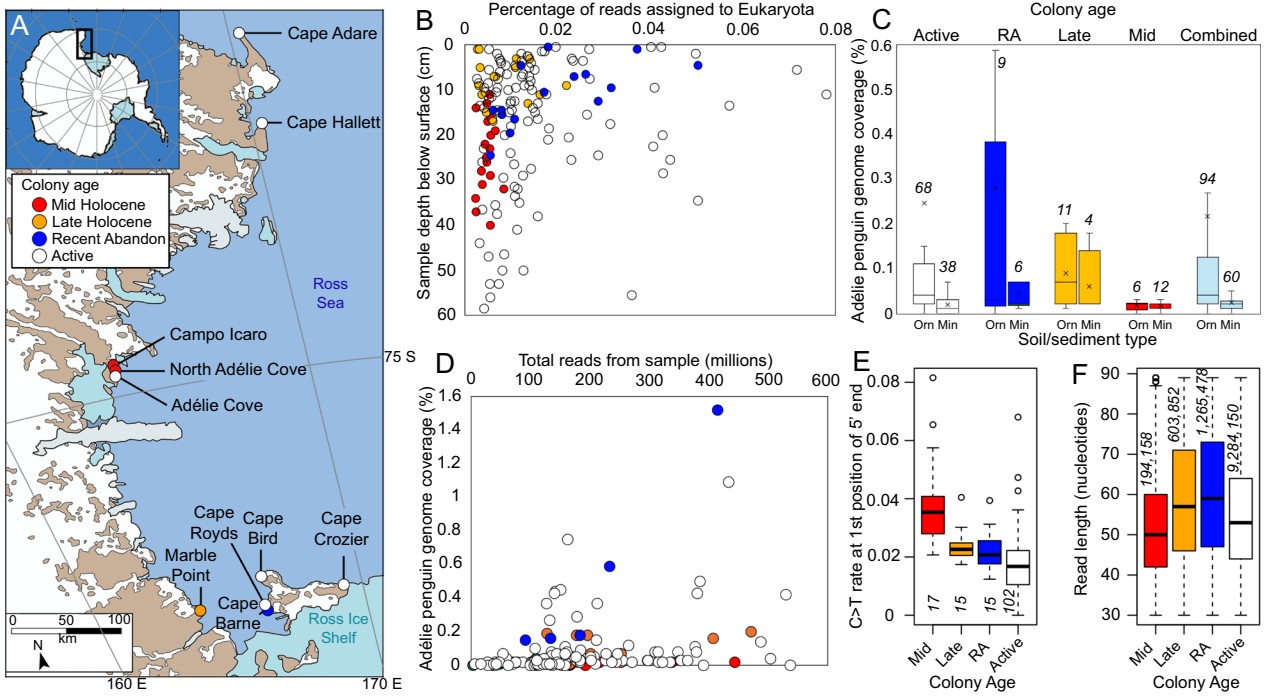

**Fig. 1 | Geographical location, colony age and metagenomics. A** Location of study sites, eastern Victoria Land bordering the western Ross Sea, Antarctica (inset in top map). **B** Percentage of sedimentary ancient DNA (*seda*DNA) reads assigned to Eukaryota and depth of sample below the ground surface. Note that an outlier (0.5 cm Cape Crozier, with 0.511% Eukaryota reads) is not shown. **C** Adélie penguin (*Pygoscelis adeliae*) genome coverage from *seda*DNA across all samples (combined) and by colony age grouped based on soil/sediment type (Orn, ornithogenic soil; Min, minerogenic sand/silt). **D** Adélie penguin genome coverage and total reads per sample. Note that an outlier (0.5 cm Cape Crozier, with 8.51% coverage) is not

shown; **E** 5' terminal cytosine deamination rates for Adélie penguin DNA. **F** DNA fragment length distributions for reads mapping to the Adélie penguin genome. Samples and groups in all plots are coloured based on colony age, as shown in the legend in Fig. 1A. The colony age grouping "RA" denotes recently abandoned colonies, while "Mid" and "Late" denote mid-Holocene and late-Holocene age colonies respectively. Italicised numbers refer to the number of samples (Fig. 1C, E) or reads (Fig. 1F; see Supplementary Data 2) in each group. Boxplot elements are median (solid line), interquartile range (grey box), minimum and maximum values excluding outliers (whiskers) and outliers (circles).

(Supplementary Table 2), although the site has been periodically reoccupied within the past few decades[33]. The Marble Point colony dates to the late Holocene, with median calibrated dates ranging from 2715–3922 years BP. One of our dates (UCIAMS 229712) is 134 radiocarbon years earlier than the previous oldest date obtained from Marble Point[34]. The colonies at Campo Icaro and North of Adélie Cove were both occupied during the mid-Holocene, with median ages ranging from 3095–5662 years BP (Supplementary Table 2). One of our dates (UCIAMS 229714) is 116 radiocarbon years earlier than the previous oldest date obtained from Campo Icaro[26].

For five of the six excavation pits where pairs of dates were obtained from upper and lower sediment layers, the 95% confidence intervals of calibrated ages overlapped (Supplementary Fig. 3). This could reflect either a short temporal occupation of the excavated nest mounds or that some eggshell fragments from upper layers might have fallen into the pits during excavation due to the unconsolidated nature of the sediments. Previous evidence suggests that the amount of time represented in each stratigraphy is likely to be relatively short, with most active colonies from the middle and northern Ross Sea having only been established < 2000 years ago, and in the southern Ross Sea within the last 1100 years[26]. However, we cannot rule out the incorporation of younger eggshells into older bulk sediment spit samples as in some cases, older dates have previously been obtained on larger materials (e.g., bones) excavated at the same colonies[26]. Only one of the six pits, Campo Icaro, had a significant difference in age across the stratigraphy (~ 2000 – 3000 years; Supplementary Fig. 3). Therefore, to account for this uncertainty, we took the conservative approach of assessing biological change through time by grouping colonies into the four age classes indicated above, and within each pit based on stratigraphic context, rather than modelling age-depth relationships.

### Sedimentary DNA characteristics

A total of 94.346 billion reads were generated from across the 156 sediment samples of which 28.439 billion (30.14%) passed quality control and human DNA filtering steps, and 1.63 billion of these (30.07%) were successfully mapped to a reference sequence (Supplementary Data 1). Of these, just over 5 million reads (0.31% of mapped reads) were assigned to Eukaryota by the LCA method (Supplementary Data 1). Within the Eukaryota reads, 76% were from Metazoa, 4.76% Viridiplantae and 4.1% Fungi (Supplementary Data 1), with the remaining 15.14% comprising protists.

The percentage of reads from each sample that were assigned to Eukaryota generally declined both with increasing age since colony abandonment and depth below the surface (Fig. 1B). As the majority of eukaryote reads identified were from taxa that live at the surface (80.6% of eukaryote reads resolved to phylum by the LCA approach were Chordata), we infer this pattern to reflect the degradation and loss of eukaryote DNA over time and concomitant increase in the relative abundance of DNA from in-situ soil microbes.

We used reads that mapped to the Adélie penguin genome for damage pattern analysis (Supplementary Figs. 4 and 5) due to the high read counts obtained for this species across samples. While we found a weak relationship between Adélie penguin genome coverage and sequencing depth, we noted that the highest genome coverage was obtained in samples from active and recently abandoned colonies, compared with late- and mid-Holocene samples (Fig. 1D). Terminal (5′) deamination rates are a widely used metric for DNA damage, reflecting the percentage of reads from a sample that exhibit a cytosine to thymine misincorporation at the first nucleotide position. For Adélie penguin DNA, terminal deamination rates increased with age (Fig. 1E), though these remained relatively low even at the oldest sites (mean < 4% of reads with damage). These low rates were supported by analysis of southern elephant seal DNA from the same samples, which exhibited mean terminal deamination rates ranging from 1 to 5.7%

across colony age groups. Comparable low deamination rates have previously been reported for sedaDNA from northern hemisphere high latitude deposits[23], demonstrating the challenges of using damage patterns for authentication of aDNA in cold climate regions. Nevertheless, terminal deamination rates were significantly greater in mid-Holocene sites compared to all other sites (Welch T-test comparing mid- to late-Holocene sites, $t = 3.55$, df = 20.529, $P = 0.001947$), and late-Holocene sites had significantly higher rates than active colonies ($t = -3.0718$, df = 27.941, $P = 0.004707$), but there was no significant difference between late-Holocene and recently abandoned colonies, or recently abandoned and active colonies. Deamination rates at the second position from the 5′ end (mean = 0.02607) were slightly higher on average than the first position (mean = 0.02153); a known artefact of the library preparation method used here[35]. At abandoned colonies, Adélie penguin DNA fragment length (Supplementary Data 2) also decreased with increasing age (Fig. 1F). Mean fragment lengths differed significantly between adjacent age groups (Welch T-test, mid- vs. late-Holocene, $t = -125.61$, df = 212853, $P < 2.2e-16$; late-Holocene vs. recently abandoned, $t = 25.731$, df = 333056, $P < 2.2e-16$; recently abandoned vs. active, $t = 174.33$, df = 424862, $P < 2.2e-16$).

### Local fauna and flora detected in sedimentary DNA

We used a combination of coassembly and direct mapping approaches against different reference sequence datasets (see methods and Supplementary Fig. 6) for assigning taxonomy to unique reads using a lowest common ancestor (LCA) approach. Unless otherwise stated, the results presented are those based on the LCA approach. Results from the mitochondrial mapping component alone are also presented where these provide additional insights.

DNA of plants, metazoans, protists and fungi was recovered from each site (Supplementary Tables 3 and 4). A high proportion of the extant local terrestrial macrofaunal and macrofloral diversity was detected, including birds, seals, soil invertebrates and mosses (Fig. 2). The most abundant metazoan genera identified by our LCA and mitochondrial analyses were local (i.e., found in the Ross Sea region of Antarctica), assigning 97.05% and 95.03% of metazoan reads to local genera respectively (Supplementary Fig. 7). Vertebrate reads were dominated by Adélie penguin, which was detected at all sites (Fig. 2). South polar skua (Stercorarius maccormicki) was the next most common vertebrate species (Fig. 2), being detected at all sites using our LCA approach and at eleven sites based on mitochondrial DNA alone (Fig. 2). South polar skua commonly nest in the vicinity of Adélie penguin colonies and are a major predator and scavenger of penguin chicks. Mitochondrial DNA of other locally occurring vertebrate species was detected more infrequently. Wilson's storm petrel (Oceanites oceanicus) and Weddell seal (Leptonychotes weddellii) were detected at two sites each, and leopard seal (Hydrurga leptonyx) and southern elephant seal (Mirounga leonina) were both detected at single sites (Fig. 2). Southern elephant seal was detected at all sites using the LCA approach (Fig. 2). The diversity of vertebrates detected is notable, as vertebrates have seldom been reported in previous Antarctic sediment or soil eDNA studies[25].

Locally occurring soil invertebrates were represented by Collembola and Eutardigrada (Fig. 2). Collembola were identified at Cape Adare 1, Cape Hallett and Marble Point, and Eutardigrada only at Cape Hallett (Fig. 2). While the LCA approach did not resolve finer scale taxa within these invertebrate groups, mitochondrial DNA from Marble Point 1 was attributed to the Antarctic collembolan species Gomphiocephalus hodgsoni, and from Cape Hallett to the tardigrade genus Ramazzottius, previously reported from the Victoria Land coast[36]. The phylum Nematoda was identified from all sites and most samples by the LCA approach but was not detected by mitochondrial DNA. While these reads may represent terrestrial soil nematodes, which are known to occur along the Victoria Land coast[37], without finer taxonomic

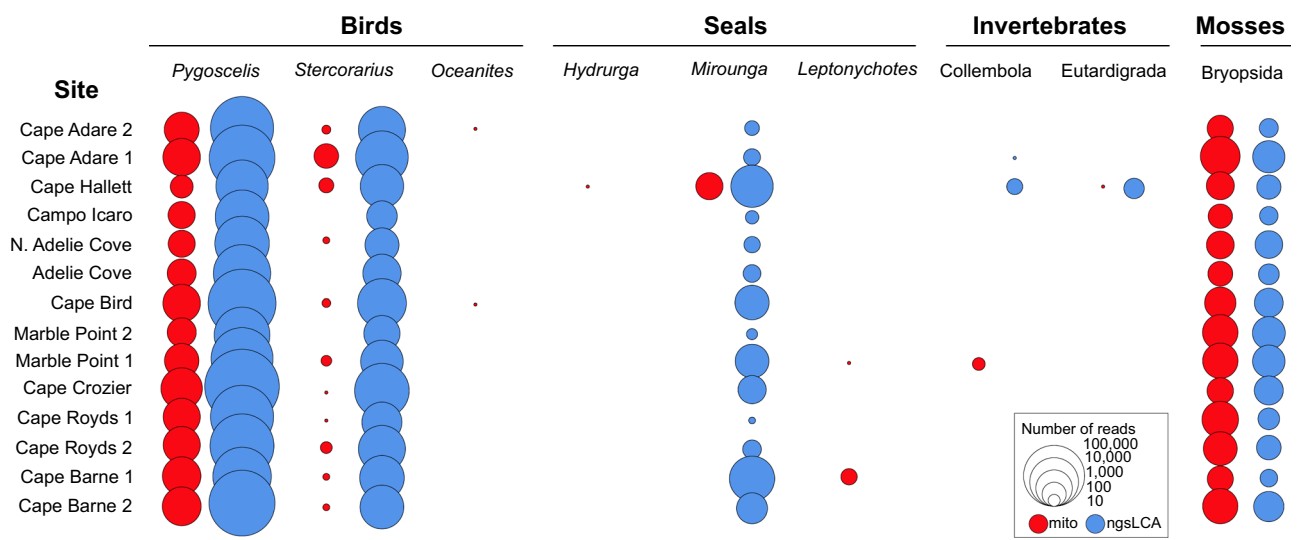

**Fig. 2 | Antarctic terrestrial flora and fauna detected by *seda*DNA.** Sites from east Victoria Land and Ross Island, Antarctica, are arranged from north (top) to south (bottom).

resolution it is not possible to distinguish between these and parasitic or free-living marine species.

Adélie penguin colonies are hotspots for moss abundance, with at least one species (*Bryum argentum*) favouring the nutrient-rich sediments at these sites[38], and this was reflected by mosses (Bryopsida) being detected at all sites.

## Spatial and temporal patterns of Adélie penguins

Within the stratigraphic context of individual pits, the relative proportion of Adélie penguin mitochondrial DNA recovered was significantly greater from ornithogenic soils compared with minerogenic sediments (sands and silts) ($t = 3.3266$, df = 139.5, $P = 0.001124$). This was also the case when nuclear DNA was included ($P < 0.0001$), with *Pygoscelis* representing a mean of 70.24% of LCA metazoan reads in samples from ornithogenic horizons, but only 55.47% in samples from minerogenic horizons. This also translated into higher coverage of the Adélie penguin genome being obtained from ornithogenic soils (Fig. 1C). It is therefore conceivable that the relative abundance of Adélie penguin DNA may correlate with local (i.e., within tens of metres of sampling site) population density. For example, lower relative abundance may reflect the presence of individual moulting birds, the presence of a colony within the general vicinity of the sampling site, or a low density of nests at the sampling site. In contrast, higher relative abundance, such as those recorded from some ornithogenic soils, may represent periods when the sampling site was located within the margins of the colony and supported a higher density of nests. *Seda*DNA sampled from multiple locations around an Adélie penguin colony could, therefore, provide a tool for tracking the contraction and expansion of colony size over time, which could be applied in conjunction with other molecular and geochemical proxies used for this purpose[32,39].

It may also be possible to gain additional insights into past colony sizes through examining signatures of genetic diversity preserved within the *seda*DNA. Significant mitochondrial haplotype diversity exists within current Adélie penguin populations[40,41]. However, analyses of nuclear microsatellite and mitochondrial haplotype data have demonstrated a lack of population structure in Adélie penguins around the Antarctic continent[42,43], indicating that despite natal philopatry and an assumed history of expansion from glacial refugia, frequent gene flow occurs between breeding colonies. Movement of individuals between colonies may be elevated during periods of difficult environmental conditions such as increased sea ice[44]. Under this

scenario, we propose that nucleotide diversity (π) within a colony should increase with colony size (i.e., the number of breeding pairs). While our data (Supplementary Data 3) show a positive linear relationship ($r^2 = 0.618$) between the maximum *seda*DNA mitochondrial π obtained from a sample within the uppermost 10 cm of sediment (i.e., the most recent past) and the current number of breeding pairs at active colonies (Fig. 3 and Supplementary Fig. 8), this is based on few data points that include replicates within colonies. The positive relationship in our data is driven by the highest π values being recovered from the two largest active colonies (Cape Adare and Cape Crozier). While this in itself is an interesting observation, increased geographic sampling will be required to test whether this relationship remains strong and consistent across a larger sample of active colonies. If so, then the relationship between colony size and mitochondrial π could provide an opportunity for estimating the approximate sizes of abandoned colonies.

Stratigraphic patterns of mitochondrial π within individual sites provide indications of changes in colony size through time. We found increases in Adélie penguin mitochondrial π towards the top of stratigraphic profiles at Cape Adare, Cape Hallett, Adélie Cove, Cape Bird and Cape Royds (pit 2) (Supplementary Fig. 9), indicative of recent population growth at these active colonies. These results are consistent both with the inferred increase in Adélie penguin populations following the end of the seal optimum around 1000 years ago[45,46] and the growth of Adélie penguin colonies within the Ross Sea since the mid-20th Century[33,47–50] that may have been partly driven by the incidental removal of prey competition resulting from commercial whaling. This stratigraphic pattern was not observed at abandoned colonies, where there was no clear or consistent trend in mitochondrial π related to sample depth (Supplementary Fig. 9).

Two major Adélie penguin lineages have previously been defined based on mitochondrial control region sequences[40]. The Antarctic lineage is widespread and found around the continent, while the Ross Sea lineage (RSL) is found mostly within the Ross Sea. This phylogeographic pattern is consistent with the post-glacial expansion of the species from two ice age refugia, and while both lineages now occur in the Ross Sea, the proportion of RSL is negatively correlated with latitude[40]. To demonstrate the potential for using *seda*DNA to study intra-specific genetic lineages in Antarctic vertebrates, we successfully resolved reads from our dataset that could be assigned to each Adélie penguin mitochondrial control region lineage (Supplementary Figs. 10 and 11; Supplementary Table 5). Higher proportions of RSL

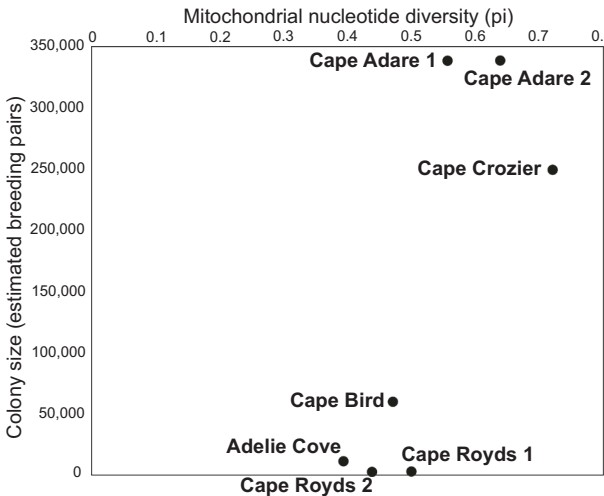

**Fig. 3 | Colony size and mitochondrial diversity.** Relationship between Adélie penguin (*Pygoscelis adeliae*) mitochondrial nucleotide diversity recovered from sedimentary DNA and colony size for active colonies. Nucleotide diversity is the maximum observed in the upper 10 cm of sediment stratigraphy.

were detected at sites north of the Drygalski Glacier (Fig. 4) (mean representation of RSL among total reads assigned to either lineage = 13.2%) compared to southern Ross Sea sites (mean = 3.1%) (two-sample $t$ test, $t = -2.0965$, df = 19.706, $P = 0.04916$). This supports the biogeographic pattern previously reported from subfossil bone samples[40], where RSL represented 18.64% of individuals south of the Drygalski Glacier and 42.86% of individuals to the north. The higher mean values reported from subfossil bones are comparable to the maximum proportions of RSL observed from *seda*DNA at individual sites in both the southern (23.8%) and northern (40%) sectors (Supplementary Table 5).

### Adélie penguin diet composition and diversity

Two major groups of Adélie penguin prey; krill (Euphasiaceae) and fish (Actinopteri), were detected in the *seda*DNA and there was an overall trend towards higher relative fish abundance in the middle Ross Sea region (Fig. 5B). This trend is consistent with the results of a recent genetic diet study of Adélie penguin diets within the Ross Sea[28] which found a greater representation of fish in the diets of Adélie penguins from the middle Ross Sea and an increasing representation of krill towards the northern Ross Sea. Our results also revealed a reciprocal increase in relative krill abundance towards the southern Ross Sea, with the exception of Cape Royds 2 where fish dominated (Fig. 5B).

Cnidarians represent a component of the Adélie penguin diet that has historically been overlooked, due to the lack of hard parts preserved in stomach content or guano deposits. However, observations using underwater cameras[51] have demonstrated the potential importance of cnidarians such as jellyfish to Adélie penguins, and this has been confirmed through a recent fecal DNA study that found jellyfish in all samples examined[52]. Our data provide evidence for spatial patterns in cnidarian consumption by Adélie penguins within the Ross Sea, with cnidarians having the highest relative abundance at sites in the northern (Cape Adare, Cape Hallett) and middle (Campo Icaro, North Adélie Cove, Adélie Cove) Ross Sea (Fig. 5a) compared to southern sites.

At most sites we observed an exact correlation between the number of fish (Actinopteri) mitochondrial reads and the number of mitochondrial reads from the Antarctic fish family Nototheniidae (Supplementary Fig. 12), indicating the fish content of these samples derived exclusively from Nototheniidae. However, at Cape Adare and Cape Hallett, a higher proportion of fish mitochondrial reads could not

be resolved to a known Antarctic fish family (Supplementary Fig. 12), indicating that other fish taxa without reference mitogenomes may be dominating the *seda*DNA signal in the northern Ross Sea. The pattern was also evident at the mid-Holocene North Adélie Cove site (Supplementary Fig. 12), but not at the modern Adélie Cove colony, indicating temporal variation in the fish taxa predated by Adélie penguins in the mid-Ross Sea. Moreover, our LCA approach failed to identify any Antarctic fish taxa from the mid-Holocene sites in the mid-Ross Sea (Supplementary Fig. 12).

The composition of Antarctic fish taxa detected varied between localities. The emerald rock cod (*Trematomus bernacchii*) was detected by both the mitochondrial and LCA methods at northern and southern Ross Sea localities (Supplementary Fig. 13). Cape Adare had a high proportion of scaly rock cod (*Trematomus loennbergii*) (Supplementary Fig. 13), a deep-water species that is most common > 300 m depth. As Adélie penguins dive to depths of < 100 m, the presence of this fish species may reflect the capture of shallower-swimming individuals by penguins feeding near the continental shelf margin adjacent to Cape Adare. The bald nototothen (*Pagothenia borchgrevinki*) was detected at Adélie Cove using both the mitochondrial and LCA methods, but otherwise was only represented at the southernmost localities (Marble Point, Cape Barne and Cape Royds) (Supplementary Fig. 13). The Antarctic silverfish (*Pleuragramma antarcticum*) was present at all localities, but its relative abundance generally declined southwards (Supplementary Fig. 13). The high relative abundance of *Pleuragramma antarcticum* and *Pagothenia borchgrevinki* in the *seda*DNA supports the observation of these two species being the most abundant notothenioid fish in an analysis of modern Adélie penguin fecal DNA from the Ross Sea[28].

The LCA method resolved two additional Antarctic fish species not identified by mitochondrial mapping. The first, South Georgia icefish (*Pseudochaenicthys georgianus*), is not known to occur locally (within the Ross Sea) and likely reflects local taxa within the same family, Channichthyidae, but which were under-represented by genomic reference sequences. For example, *Chaenodraco*, *Chionodraco*, *Chionobathyscus*, *Cryodraco* and *Chaenocephalus* were all detected in a barcoding study of modern Adélie penguin diets within the Ross Sea and together represented a relatively high proportion of fish prey detected[28]. Second, the ploughfish (*Gymnodraco acuticeps*), was most commonly detected in southern sites (Supplementary Fig. 13) and the genus is known from modern Adélie penguin diets within the Ross Sea[28]. The marbled rock cod (*Notothenia rossii*) was the rarest Antarctic fish species detected, resolved based on mitochondrial DNA from Cape Barne 2 (Supplementary Fig. 13).

The data also revealed regional changes in diet over time. Our highest density of sites was in the southern Ross Sea (Ross Island and Marble Point), where fish assemblages revealed compositional shifts over the last ca. 4000 years, characterised by a decreasing abundance of the shallow-water cryopelagic species *Pagothenia borchgrevinki*[53] and increasing abundance of *Pleuragramma antarcticum* (Fig. 5C). This pattern of change, recovered by both the mitochondrial and LCA approaches, could have been driven by declining sea ice extent and persistence in the southern Ross Sea during the late Holocene[54]. While contrasting with more northern sites[30] where *Pleuragramma antarcticum* otoliths peaked in abundance between 4000 and 2000 years BP, our pattern is consistent with a recent fecal DNA study[28] that detected a higher relative abundance of *Pleuragramma antarcticum* (56.5%) than *Pagothenia borchgrevinki* (18.21%) in modern Adélie penguin diets within the Ross Sea.

Cnidarians also appear to exhibit a temporal pattern in relative abundance, because they occurred in samples from the southern Ross Sea but were rare in the abandoned southern sites of Marble Point and Cape Barne (Fig. 5A). Moreover, our results appear to support stable dietary isotope (δ15N) data indicating increased krill consumption by modern-day Adélie penguins in the southern and

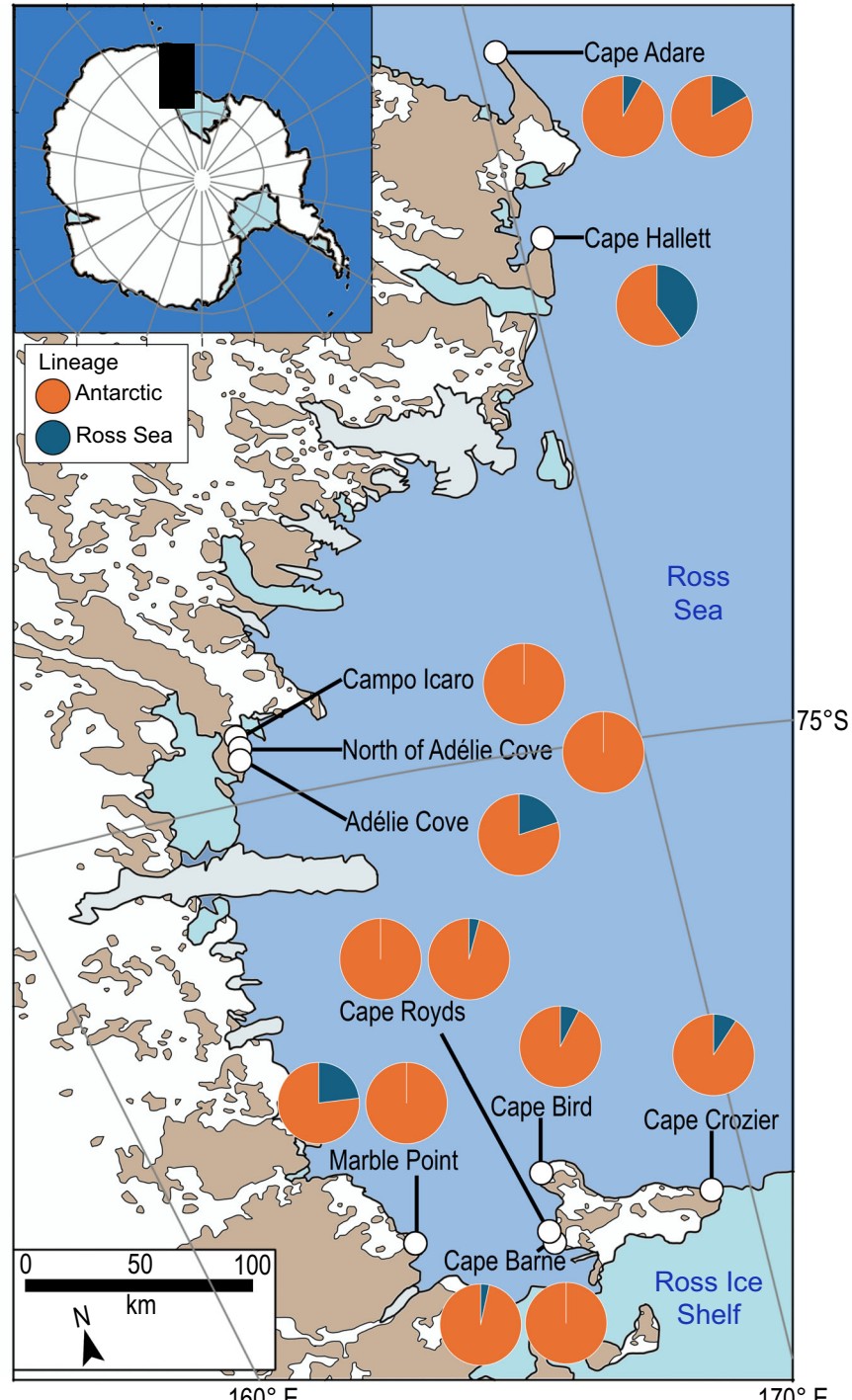

**Fig. 4 | Mitochondrial lineage proportions by locality.** Proportions of Adélie penguin Antarctic and Ross Sea mitochondrial control region lineages at each site reconstructed from *seda*DNA.

middle Ross Sea[55]. The highest relative abundances of krill (compared to cnidarians and fish) in five of our six sampled active colonies were detected in the uppermost stratigraphic sample (Cape Adare 1, Cape Adare 2, Cape Hallett, Adélie Cove, Cape Bird and Cape Crozier) (Fig. 5A).

Fish taxa detected in the *seda*DNA are consistent with those known to have been consumed by Adélie penguins in the Ross Sea during the Holocene based on a limited number of otolith studies[29,30,56]. These include *Pleuragramma antarcticum, Trematomus bernacchii, Notothenia*, Channichthyidae and *Pagothenia*. Yet, our results differ in terms of the diversity of taxa detected across

sites, indicating the potential taphonomic limitations of microscopic paleodiet techniques previously used in this region. Although krill are an important component of the Adélie penguin diet, their remains do not preserve in ornithogenic soils, and so insights into past krill predation have relied on isotopic analyses of eggshell to resolve consumed ratios of krill and fish[31]. Our detection of krill using *seda*DNA demonstrates its potential as a tool for exploring Adélie penguin paleodiets. Our inability to detect squid, another important prey item of the Adélie penguin, is an issue that could be resolved through the generation of local genomic reference sequences for this group.

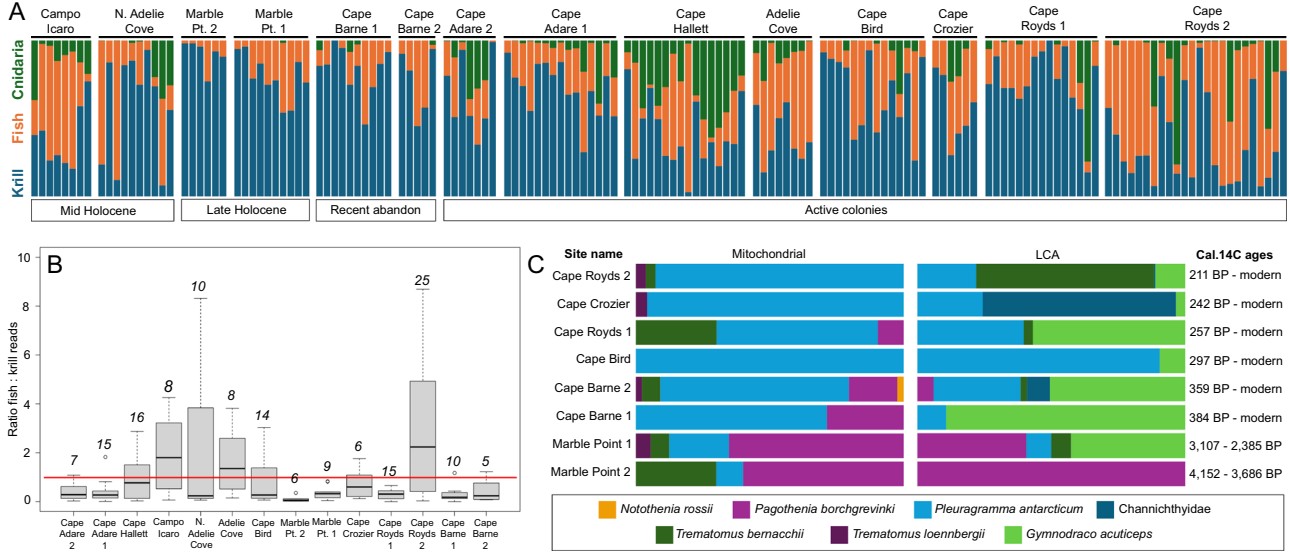

**Fig. 5 | Spatiotemporal patterns in Adélie penguin (*Pygoscelis adeliae*) diet.** **A** Relative proportions of krill (Euphausiaceae), fish (Actinopteri), and cnidarian reads for each sample based on LCA analysis. Sites are arranged by age groupings, and within these by latitude (north to south). Samples at each site are shown in stratigraphic order; (**B**) ratios of fish:krill reads for each site, arranged by latitude (north to south). Boxplot elements are median (solid line), interquartile range (grey box), minimum and maximum values excluding outliers (whiskers) and outliers (circles). Italicised numbers refer to the number of samples yielding fish and krill DNA at each site. **C** Antarctic fish assemblages recovered from southern Ross Sea sites. Sites are ordered from youngest to oldest based on calibrated radiocarbon ages obtained in this study (Supplementary Table 1).

## Colonisation history of southern elephant seals

A high relative abundance of southern elephant seal was detected in *seda*DNA from basal (> 35 cm depth) sediments at Cape Hallett (Fig. 6). A subsequent decline in the relative abundance of southern elephant seal and increase in Adélie penguin was evident from both mitochondrial and LCA approaches, and coincided with an increasing abundance of penguin eggshell and ornithogenic soil development (Fig. 6). The upper 10 cm of stratigraphy was unable to be sampled for *seda*DNA due to the lack of fine sediment, but as with the ornithogenic soil layer below also contained common penguin bones and feathers (Fig. 6). We suggest that the dominance of southern elephant seal DNA in the basal sediments at Cape Hallett is strong evidence for the former presence of a local breeding colony. While individual southern elephant seals occasionally come ashore on the Antarctic coast to rest[55,57], such sporadic visitations would likely result in DNA being detected only within a single depth horizon. The relative dominance of elephant seal DNA spanning ~ 40 cm of stratigraphy suggests a more sustained local presence of multiple individuals, such as would be expected if a breeding colony were present.

Southern elephant seal breeding colonies are currently restricted to islands within the subantarctic region of the Southern Ocean. However, the discovery of desiccated pup remains and hairs preserved in sediments indicate the species formerly moulted and bred along the Victoria Land Coast from Campbell Glacier in the north to Marble Point in the south[58]. Radiocarbon dating of elephant seal remains from the Victoria Land Coast indicates that reduced sea ice extent between ~ 2500 and 1000 years ago allowed elephant seals to breed at these southern latitudes and attain a large local population[44,45] before experiencing a local extirpation event that began in the south and progressed northwards as sea ice expanded[58,59]. The Cape Hallett *seda*DNA record provides evidence for a further late Holocene breeding colony site, the northernmost known from the Ross Sea. The exclusion of breeding elephant seals from Cape Hallett by increasing fast ice conditions would have increased the area available for breeding Adélie penguins, reflected by their increased representation in the *seda*DNA towards the top of the stratigraphy (Fig. 6).

## Marine plankton assemblages

As the base of marine food chains within the Ross Sea, plankton has the potential to directly impact populations of large predators such as Adélie penguins through bottom-up trophic processes. Moreover, plankton communities and species can serve as sensitive indicators of environmental, climatic or oceanic conditions, providing insights into how these factors may change temporally and spatially. Phyla representing key elements of Antarctic marine plankton communities were identified from the *seda*DNA including Bacillariophyta (diatoms), Chlorophyta (green algae), Rhodophyta (red algae), Heterolobosea, Ciliophora (ciliates) and Cercozoa (Fig. 7; Supplementary Table 6 and Supplementary Data 4). Spatiotemporal patterns of change in the relative proportions of phyla were evident in our data. For example, Rhodophyta was relatively common at most active colonies across the Ross Sea, except for Cape Royds (Fig. 7). Ciliophora were most represented in active colonies, especially within the mid and southern Ross Sea. With the exception of Campo Icaro, Ciliophora were notably rarer or absent in samples from abandoned colonies, including those at North Adélie Cove, Marble Point and Cape Barne (Fig. 7), indicating possible temporal variation in their relative abundance. Bacillariophyta DNA was recovered from all sites, but in the southern Ross Sea was particularly dominant at the older sites (Marble Point and Cape Barne 2) and less so at active southern colonies. *Berkeleya* represented more than half of Bacillariophyta reads (50.8%), with at least one local species, *B. adeliensis*, being cryopelagic and occurring on the underside of sea ice[60].

Stratigraphic patterns that were evident within individual sites also may indicate local temporal changes in environmental, climatic or marine conditions. For example, Bacillariophyta were absent between 26 cm–35 cm depth at Cape Hallett (Fig. 7). This corresponds to the sediment layers immediately below the ornithogenic soil horizon, and is the inferred transition zone between southern elephant seal and Adélie penguin dominance, thought to have been driven by a change in local sea ice conditions[45,46].

The presence of DNA from marine plankton demonstrates the potential for molecular archives in near-shore terrestrial

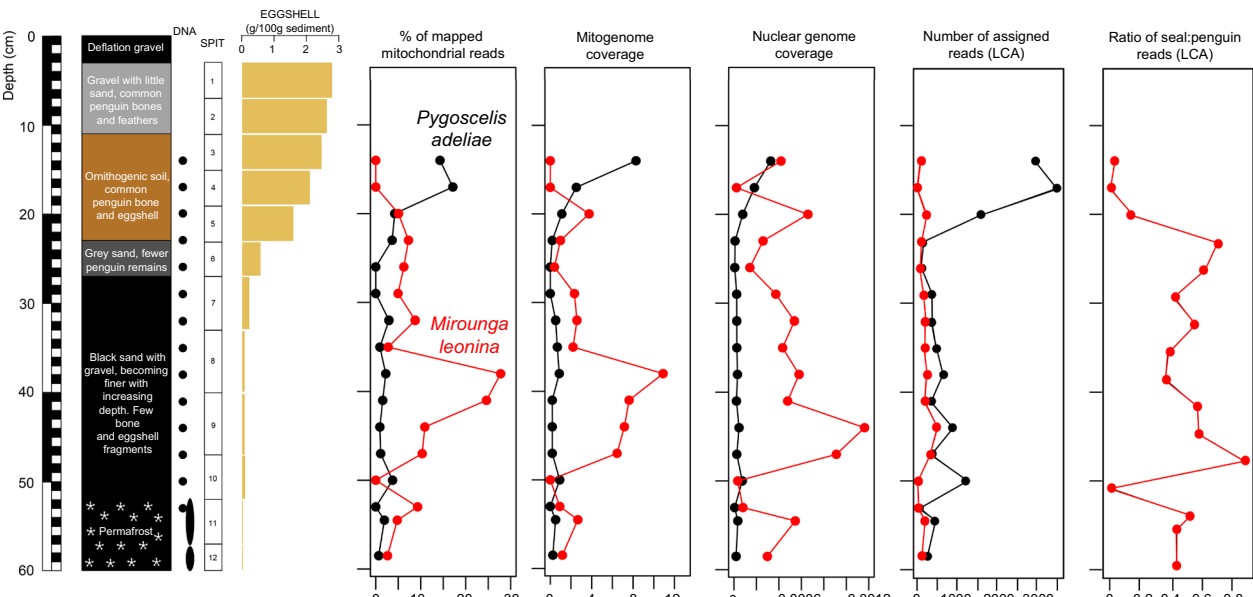

**Fig. 6 | Vertebrate fauna turnover at Cape Hallett.** *Seda*DNA records a local change from southern elephant seal (*Mirounga leonina*) to Adélie penguin (*Pygoscelis adeliae*) dominance, coincident with the formation of ornithogenic soil and an increasing abundance of sedimentary eggshell fragments.

sedimentary deposits, such as those found at Adélie penguin colonies, to be used for the study of past ocean conditions. However, as the ecological and biogeographic characteristics of Antarctic marine plankton can vary between species within the same genera[e.g. 61], and because some major groups having been undersampled locally[62], the ability to extract detailed insights from such *seda*DNA data will rely on increasing the availability of reference genomes for local indicator plankton species. Moreover, an improved understanding of the main source of this DNA, whether wind-dispersed aerosols or penguin faeces, will also be critical for understanding the potential biases involved and aiding with its interpretation.

## Prospects for Antarctic terrestrial sedaDNA

Our study provides an assessment of ancient DNA from Antarctic terrestrial (non-lacustrine) sediments and demonstrates the potential for using DNA from such records to study spatiotemporal dynamics of both terrestrial and marine biodiversity at high southern latitudes. The Cape Barne samples were especially noteworthy in being the southernmost *seda*DNA records reported to date (77° 34.8′ S), and are among the southernmost ancient DNA records yet studied (together with ancient microbial DNA from the Patriot Hills ice core, 80° 18′ S)[63]. Overall, the biological communities reconstructed from the *seda*DNA are consistent with the known spatial and temporal distributions of local species. Clear biological signals of a major local climate event, the late Holocene seal optimum, were evident in the *seda*DNA. This was represented both by changes in Adélie penguin diet composition within the southern Ross Sea and the discovery of a species turnover event between the southern elephant seal and Adélie penguin dominance at Cape Hallett. Moreover, our results demonstrate the potential for *seda*DNA as a tool for reconstructing past Adélie penguin haplotype frequencies, colony sizes and changes through time. The *seda*DNA recovered from Holocene Antarctic terrestrial sediments was well-preserved, with low terminal deamination rates comparable to those reported from northern high latitudes. This suggests that further work on such samples is warranted, with a high potential for recovering *seda*DNA records from older (pre-Holocene) sediments, such as late Pleistocene terrestrial deposits around the margins of the southern Ross Sea[26].

## Methods

### Permissions and permits

All research planning was undertaken in conjunction with Antarctica New Zealand and was subject to an environmental evaluation for activities pursuant to section 17 of the Antarctica (Environmental Protection) Act 1994. Sample collection and entry to Antarctic Specially Protected Areas - ASPAs (121, 124, 155, 157) for excavation or transiting were undertaken with the appropriate permissions and permits. Samples were imported to New Zealand under a "Permit to import laboratory specimens" (Permit No. 2018067364) granted to Landcare Research Ltd and are stored frozen at the Manaaki Whenua Landcare Research Soil Ecology & LTEL Transitional Facility, Lincoln.

### Study sites and excavations

We excavated fourteen pits at ten Adélie penguin colony sites along the western margin of the Ross Sea in January 2019 and January-February 2020 (Fig. 1). These included Cape Bird, Cape Royds (2 pits), Cape Barne (2 pits), Cape Crozier and Marble Point (2 pits) in the southern Ross Sea, Campo Icaro, Adélie Cove and North of Adélie Cove in the mid-Ross Sea, and Cape Hallett and Cape Adare (2 pits) in the northern Ross Sea. Of these, the colonies at Cape Bird, Cape Royds, Cape Crozier, Adélie Cove, Cape Hallett and Cape Adare are currently active, while Cape Barne, Marble Point, Campo Icaro and North of Adélie Cove are sites of former but now abandoned colonies. Excavations at active colonies were located on abandoned nest mounds outside the current extent of the colony. The placement of excavations at abandoned colonies was determined by digging small test holes on suspected abandoned nest mounds to confirm the presence of buried ornithogenic soil horizons. Excavated pits measured 0.09–1.0 m² in area (Supplementary Table 1). Most excavations were continued until permafrost ice was encountered, preventing further excavation with hand tools. GPS coordinates, stratigraphic descriptions and sampling details for excavation pits are provided in Supplementary Table 1 and Supplementary Data 1. Excavations were performed using trowels and sediment was removed in spits of 5 cm depth when no clear stratigraphic boundaries were present. Bones, eggshells and desiccated chick remains were recovered by sieving sediment in the field and are stored with the bulk sediment samples. Field sterility protocols that are critical for ancient DNA research were adhered to, including a

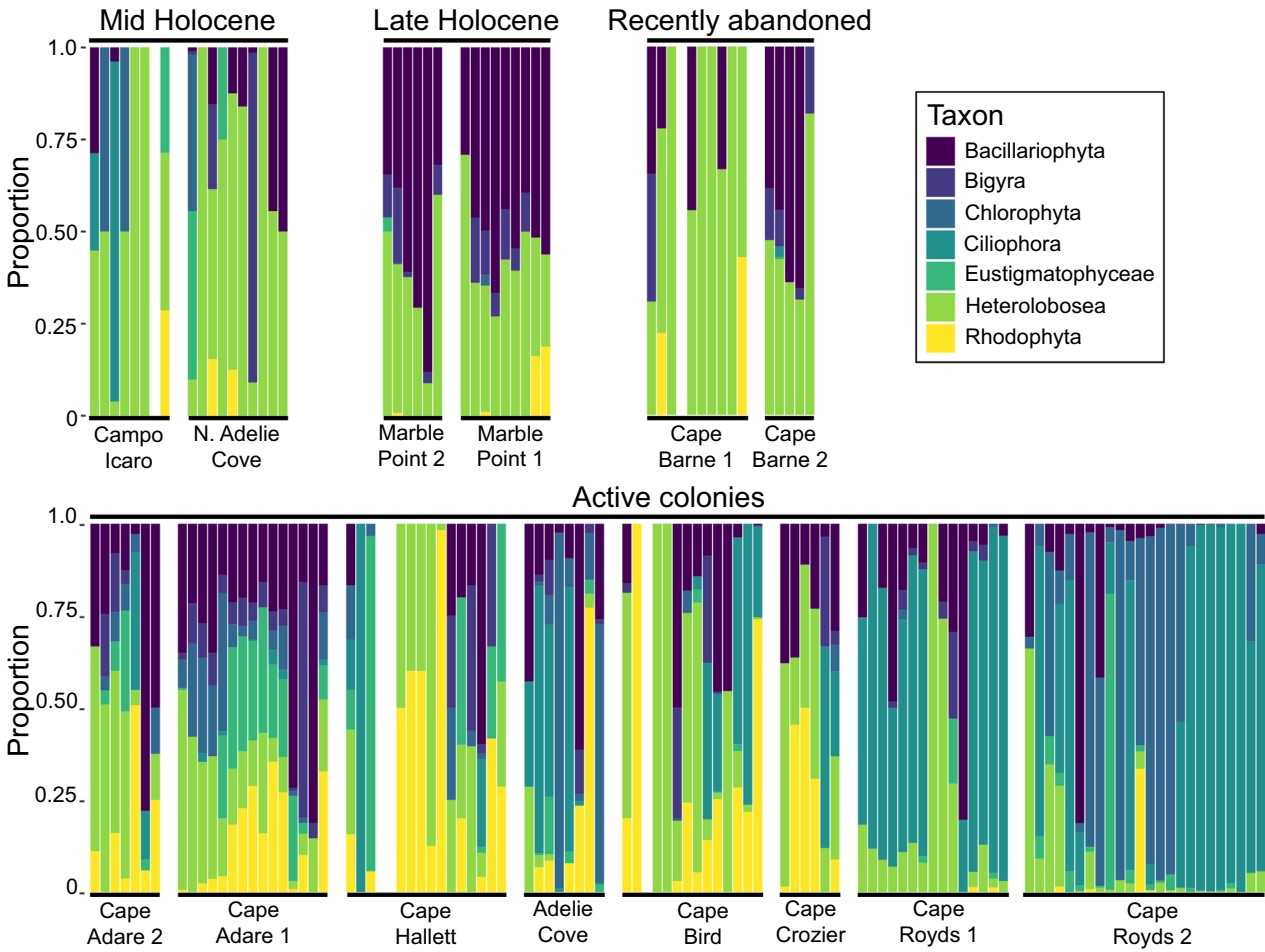

**Fig. 7 | Marine plankton assemblages.** Reconstructed using sedaDNA recovered from Adélie penguin (*Pygoscelis adeliae*) colonies, eastern Victoria Land, Antarctica. Sites are arranged by age and then within each age group by latitude (north to south). Samples within each site are arranged in order of increasing stratigraphic depth.

thorough cleaning of tools, tarpaulins and other equipment between sites using DNA AWAY™ Surface Decontaminant (Thermofisher). Sediment samples for DNA analysis were collected from freshly exposed faces at the side of pits by pushing a sterile 50 mL Falcon tube directly into the sediment or by scooping with a stainless steel spatula (cleaned between samples using DNA AWAY) and tubes remained sealed until they were subsampled within an ancient DNA laboratory.

**Penguin eggshell isolation**
Subsamples (100 g) of bulk sediment collected from each excavated spit were split into 2 × 50 mL falcon tubes (- 50 g per tube). RO water was added and the tubes were shaken to disperse the sediment. The content of each tube was tipped into nested sieves (6 mm, 1 mm and 0.25 mm mesh) and rinsed with RO water. Material collected on the 1 mm sieve was placed into a petri dish and eggshell fragments were picked out and weighed.

**Radiocarbon dating**
Between 1–15 fragments of air-dried penguin eggshell fragments (weighing 17.6 – 52.8 mg in total) were selected from individual sediment spits and radiocarbon dated at the Keck Carbon Cycle AMS facility at the University of California, Irvine, by accelerator mass spectrometry (AMS). Samples were leached 50% with dilute HCl prior to hydrolysis with 85% phosphoric acid. Sample preparation backgrounds were subtracted, based on measurements of 14C-free calcite. Results were corrected for isotopic fractionation according to the conventions of Stuiver & Polach[64], with d13C values measured on

prepared graphite using the AMS spectrometer. Small $CO_2$ yields from hydrolysis resulted in larger-than-expected uncertainties for samples UCIAMS#229695, 229708, 229712, 229714, 229715 and 229717. This could have partly been caused by the diagenetic replacement of eggshell carbonate with phosphate, a process previously observed in Adélie penguin eggshells buried in ornithogenic soils[65].

Calibration of radiocarbon dates for marine organisms in the Ross Sea region is problematic due to the high and temporally variable delta R values[66,67]. Previous calibrations of radiocarbon dates from Adélie penguin tissues and guano within the Ross Sea region have used delta R values of 750 ± 50 years[26,29] or 750 ± 0 years[31] though Hall et al.[66] suggest a long-term average delta R of 791 + /- 121 was appropriate for the area. For this study, we use the time-dependent delta R model of Hall et al.[66], where delta R varies throughout the Holocene. All calibrations were performed using the Marine20 curve[68] with the contemporaneous delta R and implemented using OxCal Online 4.4.

**DNA extractions, library construction and sequencing**
DNA extractions and library preparations were performed in a purpose-built ancient DNA laboratory at Manaaki Whenua Landcare Research, Lincoln, New Zealand. DNA was extracted from < 15 grams of each sediment sample using the Qiagen DNeasy PowerMax Soil Kit, following the manufacturer's protocol. To reduce the risk of contamination, we only undertook DNA extractions on 10 samples at any time, including an extraction negative consisting of 5 mL ddH2O.

For all 173 DNA extracts (156 sediment samples and 17 extraction negative controls), we constructed metagenomic libraries following

the Blunt-End Single-Tube (BEST 2.0) library protocol designed for modern and ancient DNA samples[35]. Library preparations were undertaken in batch sizes of between three and twenty samples at a time. Libraries were prepared using custom BGI adaptors (Supplementary Data 5). Adaptors were diluted to 0.2 μM – 2 μM depending on the sample being prepared (Supplementary Data 1). Generally, samples collected from a depth between 0 – 5 cm below the surface required 2 μM adaptors, while samples collected from > 5 cm below the surface required 0.2 μM. If substantial primer dimer was detected in libraries prepared using 2 μM adaptors, we re-built the libraries using 0.2 μM adaptors. The primers used for library amplification consisted of a common index primer used for all samples and one of thirty unique index primers used for individual libraries (Supplementary Data 5). PCR's were conducted in six batches of thirty libraries at a time, and for each subsequent batch of PCRs, the unique index primers were re-used. PCRs (25 μL) were performed in triplicate using 2 mg/mL BSA (Sigma), 1 x PCR buffer, 2 mM MgSO4, 80 μM dNTP, 1 μM common index primer, 1 μM unique index primer, 0.625 U HiFi Platinum Taq (Invitrogen) and 2 μL library on a BIO-RAD MyCycler thermal cycler with an initial denaturation of 94 °C for 3 min, followed by 30 cycles of 94 °C for 30 s, 55 °C for 30 s and 68 °C for 45 s, and a final 10 extension at 68 °C for 10 min. Libraries were checked for quality and the presence of primer dimer on a 2% agarose gel. All triplicates were then pooled together and purified using SPRI-select. DNA sequencing was undertaken at BGI-Shenzhen using the DNBSEQ-T1 platform.

## Bioinformatics

Sequence data analysis methods are described below. A diagrammatic workflow is shown in Supplementary Fig. 6 and details of scripts and commands used for each step are provided in Supplementary Code 1.

## Data processing

Several pre-processing steps were performed to ensure the quality and reliability of the sequencing data. First, the removal of sequencing adaptors, duplicates and low-quality sequences was performed using the fastp tool v.0.22.0[69] with the following parameters: "−length_required 30 --qualified_quality_phred 20 --n_base_limit 3". The fastp output reads were used in the metagenomes co-assembly step. Second, we employed the BWA aln algorithm (v0.7.17-r1188)[70] to align the fastp output reads against the human reference genome (GRCh38) to identify and remove potential human contamination from the dataset. We retained only reads that did not map to the human reference genome. bbMAP repair (v.20150602)[71] was then used to remove singletons from the extracted reads and repair the paired reads to avoid extracting errors. The repaired clean reads were used in subsequent mapping analyses.

## Database composition
### Mitochondrial datasets
**COXI database.** We downloaded all 5,608,848 entries of metazoan COXI sequences from the MetaCOXI database[72] and used this for initial taxonomic classification analysis. Though this dataset contains more species than the mitochondrial genome database (detailed below), we found that the COXI-based analysis identified fewer species than the mitochondrial genome-based analysis, and missed key taxa such as Antarctic krill. We, therefore, decided upon a whole mitochondrial genome-based analysis for this study.

**Mitochondrial genomes.** We constructed this database by integrating mitochondrial genomes from the NCBI RefSeq database (RefSeq-release206) and mitochondrial genomes of eleven penguin species[73] available on NCBI. As repetitive sequences exist within mitochondrial genomes, we performed repeat masking to identify and mask repeats using trf (v.3) with parameters "2 5 7 80 10 50 500 -d -h −m".

**Adélie penguin mitochondrial control region.** The construction of this database is detailed in the "Adélie penguin lineage identification" section of the methods.

### Nuclear and other datasets
**Vertebrate genome datasets.** We constructed vertebrate genome databases for the purposes of DNA damage estimation, fauna authentication and relative abundance estimation. Three vertebrate species were selected according to them being well represented in the mitochondrial taxonomic classification results: *Pygoscelis adeliae* (source: B10K), *Mirounga leonine* (source: Mleo_1.0) and *Stercorarius parasiticus* (source: B10K-DU-001-20). The latter was used as a congeneric for *Stercorarius maccormicki*, for which no genome was available.

**Krill.** Krill are an important dietary component for Adélie penguin. The Antarctic krill (*Euphasia superba*) is one of the major species consumed, yet has a large genome size of approximately 45 billion base pairs[74]. To optimise our computing resources, we instead utilised the Antarctic krill transcriptome (source: GFCS01.1) instead of the entire genome. Previous studies have shown that while most ancient DNA in sediment samples may originate from the nuclear genome, the mitogenome and transcriptome can still be detected.

**Nr database.** We obtained the Nr database (v.20201224) in FASTA format and the taxonomy dataset from the NCBI website and converted it into DIAMOND's (v.2.0.9.147)[75] database format using the "diamond makedb" command.

## Metagenome construction and annotation
We used MEGAhit[76] v.1.2.9, a de Bruijn graph-based algorithm tool, for metagenome co-assembly. The co-assembly method involved performing an assembly using input files derived from multiple samples. Given the limitations of computational resources, we divided the samples into seven distinct groups and performed co-assembly for each group. We then combined and clustered the seven co-assemblies to generate a final co-assembly using CD-hit with parameters "-c 0.95 -M 0 -T 2 -l 100". This final co-assembly contains 13,229,592 clusters and represents the integration of the genetic information from all the samples, providing a more comprehensive and accurate representation of the metagenomic content. The assembled contigs were aligned to the Nr database by DIAMOND with e-values ≤ 0.01. The taxonomic classification was determined using the lowest common ancestor algorithms implemented in BASTA v.1.4. The classification criteria included an alignment length > 30, identity value > 30%, and shared by at least 51% of hits. Sequences that could not be aligned or classified to any taxa were designated as unknown taxa. We also conducted a homology-based search using BLASTn v.2.12.0 against the Nt database and performed LCA annotation on the BLASTn output. However, this method resulted in a relatively low annotation rate, with only ~15% of the data being identified. Upon comparison of the two methods, the DIAMOND method demonstrated a substantially higher annotation success rate of ~56%. We retained the annotated results from the DIAMOND method for further analysis in this study.

## Mitogenome mapping and annotation
Ancient DNA is typically characterised by short fragment lengths, and previous studies[77] have shown that specific settings of BWA-aln are considered superior for aDNA data alignment than other tools. Therefore, we employed consistent settings of BWA-aln (-l 1000) for all alignment analyses conducted in our study with a minimal mapping quality of 25. In addition, we used Picard v.2.25.1 for the removal of duplicate reads to improve the accuracy and reliability of downstream analyses. For mitochondrial alignments, we selected high-quality unique reads by excluding alternative alignments, supplementary

alignments, and reads with a different reference name compared to the next read in the template ('rname = =rnext') using SAMtools (v.1.17). By implementing these stringent criteria, we were able to confidently assign a unique taxonomic classification to each mapped read. The abundance was then estimated using SAMtools and BamDeal (v.0.26). The relative abundance was normalised using a Pseudo-total sum scaling method, where counts were divided by the total sum of mapped counts in the corresponding sample.

### Genome mapping

We applied BWA-aln to map the clean read data onto the constructed animal genome reference datasets, including genomes of *Pygoscelis adeliae, Mirounga leonine, Stercorarius parasiticus*, the transcriptome of *Euphausia superba*, and the final co-assembly dataset separately. PCR duplicates and low-quality mapping alignments were removed. This generated 5 bam files for each sample.

### LCA-based taxonomic assignment

Taxonomic profiling was conducted using ngsLCA v.1.0.0 following Wang et al.[78]. Firstly, for all custom genome references, we assigned a pseudo NCBI-styled accession number to each non-NCBI-standard entry based on its taxonomic classification and made modifications to the accession2taxid file by adding new annotations that included the pseudo identifier and taxaID. Next, we merged all six mapping alignments (the mitochondrial genome alignment and genome alignments) into a single sorted bam file for each sample using the "samtools merge" command. Subsequently, we performed a lowest common ancestor on all merged alignments of a given read that have between 0 and 10 mismatches and a similarity score ranging from 95% to 100%. Thus, ngsLCA assigned the lowest common ancestor of all hit taxa to each of the reads with multiple alignments.

We utilised the ngsLCA R package to generate a complete taxonomic profile by merging all ngsLCA result files and performing decontamination based on negative controls. Specifically, we combined the taxa profile for controls using three thresholds in "ngsLCA_filter": (1) a minimum read number of 3 to confirm the presence of a taxon in each sample, (2) the default minimum read percentage of 0, and (3) a minimum sum of reads across all samples of 50. To refine the taxa profile, we excluded known fauna species of Antarctica from the list of negative species and filtered out reads assigned to *Homo sapiens*, primates, or species detected in the controls to avoid contaminations for the merged samples. In addition, we applied a minimum sum of reads across all samples of 20 to obtain the final taxa profile. We then grouped merged taxa profiles into different taxa units using "ngsLCA_group", including Eukaryota, Viruses, Archaea, Bacteria, Fungi, Viridiplantae and Metazoa and classified taxa to taxonomic ranks using "ngsLCA_rank". Reads and taxa numbers were counted using "ngsLCA_count".

### Damage analyses

To validate DNA damage, we employed mapDamage (v.2.1.0)[79] on reads mapped to Adélie penguin. Initially, a cross-checking validation step was performed to eliminate non-target species reads from the single genome alignments based on the ngsLCA-based taxonomic profiling results using "samtools view". Following that, we conducted mapDamage analysis on the filtered BAM files to evaluate the damage patterns exhibited by the target species in each sample. During this analysis, we disregarded reference sequence names when tabulating reads by utilising the "–merge-reference-sequences" option. In addition, we calculated the read length and count for each sample to provide comprehensive information on the read characteristics.

### Adélie penguin lineage identification

Whole mitogenome reconstruction was not used here due to the likely presence of multiple individuals within each sample. Instead, we used a haplotype-based approach to assign individual reads to lineages. We downloaded hundreds of control region sequences from previous studies, including control region sequences from both Antarctic and Ross sea lineages[40], Antarctic lineage sequences[80], control region sequences from Adélie penguin mitogenomes[81,82] and the outgroups *Pygoscelis antarcticus* and *Pygoscelis papua*[81,83] from NCBI to determine the lineage distribution of the online dataset. We aligned all control region sequences using MAFFT v.7.487 and inferred the phylogenetic tree using the maximum likelihood-based method IQ-Tree with outgroup rooting and 5000 ultrafast bootstrap replicates after model testing. Additionally, a minimum-spanning haplotype network was constructed using PopART[84] v.3. Although the Antarctic lineage (A lineage) appeared as a paraphyletic group in the rooted tree, we were able to differentiate two distinct groups based on the topology and branch lengths observed in the tree and their distribution within the haplotype network. In total, 134 sequences were identified as Ross Sea lineage (RS lineage). To minimise potential bias, a random subset of 134 sequences from the A lineage was selected from the nonredundant sequences for database construction. Each sequence was labelled with either A_ or RS_ based on the cluster information provided in the tree to facilitate further analysis. Finally, the control region database was constructed using Makeblastdb (v.2.11.0). We applied the Pathfinder workflow[85], but there were insufficient shared informative loci to allow the identification of mitogenome haplotypes across samples. Instead, we used a Pesudo Lowest Common Ancestor (PLCA) method for penguin lineage identification. First, paired-end reads were aligned to the Adélie penguin mitogenome using the tool BWA (v.0.7.17-r1188) with similar parameters and the paired-end reads were extracted from the alignment results using SAMtools. Subsequently, a BLASTn alignment was performed on the mapped reads against the reference control region sequences BLASTn database. The alignment analysis was conducted with an *e*-value cut-off of 1e-5, and hits were filtered with an identity cut-off of 95% and a minimum unique reference number of 3. We then used a PLCA method to assign the read pair to either the A lineage (if >70% of the hits belonged to the A lineage), the RS lineage (if >70% of the hits belonged to the RS lineage), or as undetermined (if ≤70% A and ≤70% RS). Finally, the proportion of lineages identified from each sample was calculated by determining the relative reads abundance of A and RS lineages based on the assigned reads.

### Adélie penguin nucleotide diversity

In assessing nucleotide diversity (π) across our samples, the calculation was adjusted to decrease the influence of missing data. First, SNP calling was performed using the Adélie penguin mitochondrial genome as a reference to identify SNPs, while excluding indels and ensuring a minimum read depth of 2 and a mapping quality score of at least 20. C > T and G > A transitions within 10 nucleotides of the ends of reads were recoded as N and quality score 0 to remove any potential influence of DNA damage (Supplementary Figs. 4 & 5) on nucleotide diversity rates. Subsequently, for each sample or each locality, π for each site was calculated using vcftools with "−site-pi" command. Then, we utilised the awk function to compute the average π value for each sample or each locality: the π value were summed, and the average was calculated by dividing by the number of sites with data (the adjusted sample size). We also calculated the average π for each sample and each locality specifically for the control region.

### Reporting summary

Further information on research design is available in the Nature Portfolio Reporting Summary linked to this article.

## Data availability

The source sequencing data generated in this study, with tag look-up information, have been deposited in the CNSA of the CNGBdb

database under the accession number CNP0002256 and in the NCBI as BioProject PRJNA1202586. The locations and depths of samples from which these data were generated are provided in Supplementary Data 1.

## Code availability

Published tools were used as described in the methods section, and scripts as provided in Supplementary Code 1.

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

## Acknowledgements

We are indebted to the substantial logistical support provided over two summer seasons by Antarctica New Zealand and the Italian

National Antarctic Research Programme in supporting the field project K083-D. We especially acknowledge Chris Long and Steve Grieves (Antarctica New Zealand) for their field safety support, Karen Boot and Nic Bolstridge for laboratory support, and Dr. Tianming Lan for his valuable contributions to the data analysis. This research is part of the Ross Sea Region Research and Monitoring Programme (Ross-RAMP) funded by the New Zealand Ministry for Business, Innovation and Employment (grant C01X1710). G.Z. was funded by a Villum Investigator Grant (no. 25900) from The Villum Foundation.

## Author contributions

Conceptualisation: J.R.W., C.Z., T.L.C., D.P.A., P.O.B.L. and G.Z.; Methodology: J.R.W., T.L.C. and C.Z.; Sample collection: J.R.W., T.L.C. and M.C.; Investigation: J.R.W., C.Z., T.L.C., S.T. and J.R.S.; Formal analysis: J.R.W., C.Z., S.T., X.X., X.L., S.L., M.L. Q.L.; Writing—original draft: JRW, CZ; Writing—review and editing: all authors have reviewed and approved the manuscript; Visualisation: J.R.W. and C.Z.; Funding acquisition: J.R.W., D.P.A., P.O.B.L. and G.Z.

## Competing interests

The authors declare no competing interests.

## Additional information

[1]Australian Centre for Ancient DNA, School of Biological Sciences, North Terrace Campus, University of Adelaide, Adelaide, South Australia, Australia. [2]Environment Institute, University of Adelaide, North Terrace Campus, Adelaide, South Australia, Australia. [3]BGI Research, Wuhan, China. [4]State Key Laboratory of Genome and Multi-omics Technologies, BGI Research, Shenzhen, China. [5]Manaaki Whenua Landcare Research, PO Box 69040 Lincoln, New Zealand. [6]College of Life Sciences, University of Chinese Academy of Sciences, Beijing, China. [7]Guangdong Provincial Key Laboratory of Marine Biotechnology, Institute of Marine Sciences, Shantou University, Shantou, China. [8]College of Life Sciences, Wuhan University, Wuhan, China. [9]Department of Earth System Science, University of California-Irvine, Irvine, California, USA. [10]Center for Evolutionary & Organismal Biology and Women's Hospital at Zhejiang University School of Medicine, and Liangzhu Laboratory, Zhejiang University Medical Center, Hangzhou, China. [11]Liangzhu Laboratory, Zhejiang University Medical Center, Hangzhou, China. [12]Villum Center for Biodiversity Genomics, Section for Ecology and Evolution, Department of Biology, University of Copenhagen, Copenhagen, Denmark. [13]These authors contributed equally: Jamie R. Wood, Chengran Zhou. ✉e-mail: jamie.wood@adelaide.edu.au; guojiezhang@zju.edu.cn

