## [Peer Review file · Nature Communications]

Sedimentary DNA insights into Holocene Adélie penguin (*Pygoscelis adeliae*) populations and ecology in the Ross Sea, Antarctica

Corresponding Author: Dr Jamie Wood

Version 0:

Reviewer comments:

Reviewer #1

(Remarks to the Author)
attached.

Reviewer #2

(Remarks to the Author)
Comments on NCOMMS-24-51046-T
Spatiotemporal dynamics of Antarctic biodiversity revealed by terrestrial sedimentary ancient DNA

Wood et al present shotgun data of 156 samples from 14 sites that represent either active, recently abandoned, Late Holocene or Early Holocene Adélie penguin colonies. All samples yield sufficient number of reads for interpretation of the biological content. The data analyses and presentation are very good. This is among the few studies that provide a good evidence between a carnivore and its prey through time and space, and I am sure it will be greatly appreciated by the research community. Especially, I liked the data presented in figure 4-6, where diet and marine plankton is discussed - this is the strength of the paper.

Main comments:

I am not convinced that Figure 3A shows a correlation between current colony size and mitochondrial nucleotide diversity. There are only seven points, and no statistics for the correlation is presented. Further, there are only five points in terms of colony size, and for those two colonies where multiple cores are taken, they either show similar mitochondrial diversity (Cape Adare) or different levels of mitochondrial diversity at Cape Royds. The latter is described as "minor discrepancy" although they are 0.292 and 0.5 compared to the total range of active colonies that are approximate 0.26-0.75 (read from figure 3, I could not find the number anywhere).

What is the deamination rate for the other detected taxa? Are these rates comparable to that observed for the penguins? The deamination rate for the penguins is rather low, which could be due the cold environment and better preservation. However, in the stratigraphy and chronology section it was mentioned that some younger eggshells might be incorporated into the older sediments, possibly introducing fresher genetic material and affecting the observed deamination rates. The (hopefully) independent rates for other taxa could provide some certainty against the possibility of reworked material.

For the metagenome construction the authors pooled multiple samples together. 1) Which samples were grouped together? This isn't mentioned in the main text or any of the supplementary tables that I've seen. 2) Because the samples were pooled together, and subsequently clustered into one final data, how were the metagenomic signals obtained on a per sample basis? An easier approach would be to pull all samples together, generate contigs with MEGAHIT, and use BWA to map the reads for sample, similar to the mitogenome and genome analysis.

Minor comments:

- Although the deamination rates are low, were they considered when calculating the nucleotide diversities for the penguin colonies? So, excluding any SNP calls based on bases near the end of reads, or alternative, excluding all transitions to be even more conservative.
- The range in nucleotide difference within the small colonies is larger (0.0833-0.5-ish) than the difference between the small and large colonies (both are 0.5-ish). How well does the nucleotide difference actually predict colony size?
- The result paragraphs describing the penguin diet are rather long-winded and can benefit from being edited down a bit.
- Which and how many libraries suffered from the high primer-dimer presence and had to be re-prepped with a lower adapter concentration? This should be provided in Table S2.
- CD-hit and DIAMOND are tools designed for protein clustering and alignments. Why were these chosen over tools that are designed for non-coding data?
- Figure 1C, E and F: provide the number of samples for each category (Mid, Late, RA, Active, as well as the Soil types).
- Figure 6: 17 categories in a stacked bar plot, many with very similar colours, might be a bit too much. Consider cutting some of the categories and provide the raw data in a supplementary table.

See further comments in attached pdf files.

Reviewer #3

(Remarks to the Author)

Thank you for the opportunity to review the manuscript "Spatiotemporal dynamics of Antarctic biodiversity revealed by terrestrial sedimentary ancient DNA".

I am not a specialist in DNA or sedaDNA, so my review is only concerning the radiocarbon dating and age modeling.

It is difficult to assess the degree of sediment reworking, which is also a concern raised by the authors. It would strengthen the dating results significantly, and possibly eliminate potential of sediment reworking, if the authors could provide more than one radiocarbon date from each layer. However, I do understand that this might not be practically possible if the sample material is limited.

I suggest that the presentation of dating results should be made more clear, for example by annotating radiocarbon dating results on each stratigraphic profile in Extended Data Fig. 2.

As a general remark, I think the authors present a robust dataset and I appreciate that the authors take a conservative approach to dating results and use age groups rather than age-depth models.

I recommend that the manuscript can be published with the suggested revisions.

Version 1:

Reviewer comments:

Reviewer #1

(Remarks to the Author)

I am reviewing this paper for the second time and appreciate the authors edits and additions to the text to clarify various points. I am still finding it extremely interesting the vast information that can be obtained from sedaDNA. As I noted in my last review, I am not able to comment on any of the sediment collection, radiocarbon dating, or DNA extraction methods as this is not within my expertise. I have no further comments on the manuscript.

(Remarks on code availability)

Reviewer #2

(Remarks to the Author)

The revised manuscript looks good.

(Remarks on code availability)

The revised manuscript looks good.

Reviewer 1

This is an interesting approach to understanding long-term changes in species abundance/presence and biodiversity. SedaDNA has received little attention at southern high latitudes. This study takes an impressive number of sediment samples in the Ross Sea region to determine temporal and spatial patterns in biodiversity.

I find the title to be a bit broad and I was expecting more biodiversity metrics/indices and comparisons given “biodiversity” was in the title. The study is mostly centered around Adélie penguins as the samples were collected at Adélie penguin colonies but some of the results are disjointed and read as a report rather than an interesting integration of detecting multiple species across the food web. I would be interested to see more results/statistics that compare/integrate the various species detections. I do find that the method seems promising, the results are interesting generally and specifically in understanding Adélie penguin diet/populations but sometimes the results are not totally interpretable to someone outside of this field.

We acknowledge the reviewer’s point in regard to the article’s focus; i.e. that rather than being a general overview of biodiversity (i.e., including microbes) with associated metrics, the aim is to explore in detail the flora and fauna associated with Adélie penguin colonies. We have therefore decided to update the article’s title to reflect this, which is now “Sedimentary DNA insights into Holocene Adélie penguin (*Pygoscelis adeliae*) populations and ecology in the Ross Sea, Antarctica”. We have also attempted to add further explanation to technical points specific to the ancient DNA field within the main text, in order to make these easier to follow for readers outside the field. We thank the reviewer for pointing these out in their comments below.

I do want to note I am not able to comment on any of the sediment collection, radiocarbon dating, or DNA extraction methods as this is not within my expertise.

In methods “(2) Antarctic krill:”. Crystal krill is also an important dietary item.

Our intention here was to focus on an important diet species for which there was a large amount of reference sequence data (i.e. a genome or transcriptome). A genome or transcriptome assembly is not currently available for crystal krill. But the reviewer makes a very useful point. Because crystal krill is congeneric with Antarctic krill, there is potential that for some reads we may not be able to discriminate between the two species based on a sole reference sequence. This is why in our results and discussion we only referred to ‘krill’ collectively, but we have updated the methods section text to also reflect this.

Figure 1: define “RA”. Describe “combined” in “c”? Are B/D totally necessary for the main text? Within the caption, you can you give time frames for Mid/Late/RA?

We have clarified the term “combined” within the caption for Fig. 1c, i.e.: “c, Adélie penguin (*Pygoscelis adeliae*) genome coverage from sedaDNA across all samples (combined) and by colony age grouped based on soil/sediment type (Orn, ornithogenic soil; Min, minerogenic sand/silt)”. We also now define the age classes within the figure caption, i.e.: “The colony age grouping “RA” denotes recently abandoned colonies, while “Mid” and “Late” denote mid-Holocene and late-Holocene age colonies respectively.”.

We believe that B/D are necessary and important in terms of understanding the overall dataset and taxonomic representation within the samples. We refer to B in the “Antarctic terrestrial sedaDNA characteristics” section of the results, as it clearly demonstrates the proportion of sequences from eukaryotes within a sample declined with increasing depth

below the surface. D is important in demonstrating that: (1) for most samples only a very small proportion (<1.6%) of the Adélie penguin genome could be retrieved; (2) this wasn't strongly correlated with sequencing depth, and; (3) there was a relationship with age, because the highest genome coverage was obtained in samples from active and recently abandoned colonies, compared with late- and mid-Holocene samples. Although we hadn't previously referred to Fig. 1D specifically within the manuscript text, we have done so now.

Can you describe what Terminal (5') deamination rates means?

We have added the following description to the manuscript text: "Terminal (5') deamination rates are a widely used metric for DNA damage, reflecting the percentage of reads from a sample that exhibit a cytosine to thymine misincorporation at the first nucleotide position."

Fig 1E. Interesting that mid is lower than late/RA. You say "Mean fragment lengths differed significantly between all age groups" but the stats don't include a comparison between all groups.

Thank you for pointing this out. We have corrected the wording to "Mean fragment lengths differed significantly between adjacent age groups".

No mention of collembola and eutardigrada in the text (Fig. 2).

We have added the following text to the manuscript: "Locally occurring soil invertebrates were represented by Collembola and Eutardigrada (Fig. 2). Collembola were identified at Cape Adare 1, Cape Hallett and Marble Point, and Eutardigrada only at Cape Hallett (Fig. 2). While the LCA approach did not resolve finer scale taxa within these invertebrate groups, mitochondrial DNA from Marble Point 1 was attributed to the Antarctic collembolan species *Gomphiocephalus hodgsoni*, and from Cape Hallett to the tardigrade genus *Ramazzottius*, previously reported from the Victoria Land coast. The phylum Nematoda was identified from all sites and most samples by the LCA approach but was not detected by mitochondrial DNA. While these reads may represent terrestrial soil nematodes, which are known to occur along the Victoria Land coast, without finer taxonomic resolution it is not possible to distinguish between these and parasitic or free-living marine species."

Fig 2. "Southern elephant seal was detected at all sites using the LCA approach, but these results could include other seal species". Can you explain a bit more? Is it that there is uncertainty around the presence or abundance of elephant seals detected, or something else?

We agree that this is an unclear statement and on review think our original interpretation is speculative. We have revised this sentence to simply read "Southern elephant seal was detected at all sites using the LCA approach (Fig. 2)".

Interesting about the cnidarian results. I see the mention of squid but what about octopus? These are occasionally seen in Adélie diets. Did you find any beaks from either species in the sediment samples?

A very small number of reads were resolved as Octopoda by the LCA approach; in just 6 samples where 5 were represented by <10 reads. As such, and due to the absence of squid in the LCA results, we feel that no weight can be given to these results in terms of interpreting diet.

We did perform a limited amount of faunal remain identification on associated sediments and identified the presence of squid beaks in addition to fish otoliths. However, such analyses have previously been performed at the same study sites by other researchers,

and their results demonstrating the presence of fish and squid remains in the sediments at these sites have already been published, for example:

- Polito, M., Emslie, S. D. & Walker, W. A 1000-year record of Adélie penguin diets in the southern Ross Sea. *Antarct. Sci.* **14**, 327-332 (2002).
- Lorenzini, S., Olmastroni, S., Pezzo, F., Salvatore, M. C. & Baroni, C. Holocene Adélie penguin diet in Victoria Land, Antarctica. *Polar Biol.* **32**, 1077-1086 (2009).
- Lorenzini, S., Baroni, C., Baneschi, I., Salvatore, M. C., Fallick, A. E. & Hall, B. L. Adélie penguin dietary remains reveal Holocene environmental changes in the western Ross Sea (Antarctica). *Palaeogeog. Paleoclim. Palaeoecol.* **395**, 21-28 (2014).

The plankton figure/results seem unrelated to the main penguin story. Are some plankton ice-associated or relevant to fish vs. krill diet?

Both are relevant in this context. We have added the following text to the beginning of this paragraph to make the link between penguins and plankton explicit: “As the base of marine food chains within the Ross Sea, plankton have the potential to directly impact populations of large predators such as Adélie penguins through bottom-up trophic processes. Moreover, plankton communities and species can serve as sensitive indicators of environmental, climatic or oceanic conditions, providing insights into how these factors may change temporally and spatially.”

Reviewer 2

Wood et al present shotgun data of 156 samples from 14 sites that represent either active, recently abandoned, Late Holocene or Early Holocene Adélie penguin colonies. All samples yield sufficient number of reads for interpretation of the biological content. The data analyses and presentation are very good. This is among the few studies that provide a good evidence between a carnivore and its prey through time and space, and I am sure it will be greatly appreciated by the research community. Especially, I liked the data presented in figure 4-6, where diet and marine plankton is discussed - this is the strength of the paper.

Main comments:

I am not convinced that Figure 3A shows a correlation between current colony size and mitochondrial nucleotide diversity. There are only seven points, and no statistics for the correlation is presented. Further, there are only five points in terms of colony size, and for those two colonies where multiple cores are taken, they either show similar mitochondrial diversity (Cape Adare) or different levels of mitochondrial diversity at Cape Royds. The latter is described as “minor discrepancy” although they are 0.292 and 0.5 compared to the total range of active colonies that are approximate 0.26-0.75 (read from figure 3, I could not find the number anywhere).

We appreciate the reviewer’s comments on this result, and acknowledge that the number of data points is extremely limited on which to make robust interpretations about correlations between colony size and nucleotide diversity. We believe that having obtained the highest calculated nucleotide diversity rates from the two largest colonies is an interesting observation that is worth mentioning, however, we acknowledge that the manuscript currently places too much emphasis on the correlation between these variables, and so we have: (1) rewritten the relevant sections using more careful language to address this, and (2) propose further work to explore this relationship in more detail. We have also added the relevant raw data that underpins Fig. 3 into a new supplementary table (Table S6).

What is the deamination rate for the other detected taxa? Are these rates comparable to that observed for the penguins? The deamination rate for the penguins is rather low, which could be due to the cold environment and better preservation. However, in the stratigraphy and chronology section it was mentioned that some younger eggshells might be incorporated into the older sediments, possibly introducing fresher genetic material and affecting the observed deamination rates. The (hopefully) independent rates for other taxa could provide some certainty against the possibility of reworked material.

We apologise for any confusion regarding the phrase “incorporation of younger eggshell into older sediment layers”. We did not intend to convey that this was due to sediment reworking or that this was a pervasive issue with the stratigraphic integrity of the sites, but simply that some eggshell fragments from upper layers (where eggshell was abundant) could have fallen down the sides of the pit while it was being excavated due to the unconsolidated nature of the sediments. We have now clarified this point in the manuscript. As we suggest, this could have implications for the radiocarbon dates from deeper layers, because these were obtained on rare eggshell fragments that were sieved from bulk sediments collected from spits during excavations. However, this would not have impacted the DNA results (i.e. penguin deamination rates), because great care was taken to clean back the face of the pit wall to expose *in situ* sediments prior to sampling for sedaDNA.

Adelie penguin DNA was used to calculate overall deamination rates due to the large number of reads, and the fact that their DNA originates at the surface layers of active colonies (compared to other organisms that might be in the soil, or still occur at abandoned colonies). However, we can confirm that 5' deamination rates in southern elephant seal reads were comparable. We have added this information to the manuscript: “These low rates were supported by analysis of southern elephant seal DNA from the same samples, which exhibited mean terminal deamination rates ranging from 1 to 5.7% across colony age groups.”.

For the metagenome construction the authors pooled multiple samples together. 1) Which samples were grouped together? This isn't mentioned in the main text or any of the supplementary tables that I've seen. 2) Because the samples were pooled together, and subsequently clustered into one final data, how were the metagenomic signals obtained on a per sample basis? An easier approach would be to pull all samples together, generate contigs with MEGAHIT, and use BWA to map the reads for sample, similar to the mitogenome and genome analysis.

(1) We have added the requested group information to the revised Table S3.

(2) Given the large number of sequences involved and limitations of computational resources, we could not process all samples together to generate contigs with MEGAHIT. Therefore, we divided the samples into groups and performed co-assembly for each group. After combining and clustering the co-assemblies to create a comprehensive set of non-redundant contigs, we used BWA to map the reads for each sample back to these non-redundant assembled contigs. This mapping allowed us to calculate the abundance of each contig in each sample, providing an accurate measure of the metagenomic signals on a per sample basis. Additionally, we employed ngsLCA (this tool assigns taxonomic labels to sequences based on the lowest common ancestor principle, which helps in accurately identifying the taxonomic origin of the sequences) and multiple databases to further refine

and quantify the abundance, ensuring a robust and comprehensive analysis of the metagenomic data.

Minor comments:

Although the deamination rates are low, were they considered when calculating the nucleotide diversities for the penguin colonies? So, excluding any SNP calls based on bases near the end of reads, or alternative, excluding all transitions to be even more conservative.

In the original manuscript we did not apply specific filtering for potential deamination artifacts when calculating the nucleotide diversities. To address this, we have rerun our analysis after excluding specific transitions (C->T and G->A, which are the most common deamination-induced changes in ancient DNA) within 10 nucleotides from the end of reads. This distance was guided by our DNA damage analyses (Extended Data Fig. 4) which found elevated rates of transitions generally occurred within 10 nucleotides from the end of reads. As expected due to the low damage rates, the results of this new analysis showed only minor differences with those of the original. However, we have updated our methods and results to reflect the revised analysis accounting for the exclusion of potential DNA damage.

The range in nucleotide difference within the small colonies is larger (0.0833-0.5-ish) than the difference between the small and large colonies (both are 0.5-ish). How well does the nucleotide difference actually predict colony size?

Please see our earlier response in regard to the relationship between nucleotide diversity and colony size.

The result paragraphs describing the penguin diet are rather long-winded and can benefit from being edited down a bit.

The detailed content of these paragraphs is important, as the insights about Adelie penguin diets are a strength of the paper and will be of great interest to ecologists and penguin researchers working in the Ross Sea. However, where possible we have attempted to tighten the text through rephrasing and removing any superfluous words.

Which and how many libraries suffered from the high primer-dimer presence and had to be re-prepped with a lower adapter concentration? This should be provided in Table S2.

The requested information has now been added to Table S3.

CD-hit and DIAMOND are tools designed for protein clustering and alignments. Why were these chosen over tools that are designed for non-coding data?

CD-hit and DIAMOND were chosen over tools designed for our nucleotide data due to their high performance, speed and accuracy in handling DNA sequences.

(1) CD-hit: CD-hit was initially designed for protein clustering, but it has since been adapted to handle nucleotide sequences as well. Its high performance and efficiency in clustering large datasets make it a good choice for both coding and non-coding nucleotide sequence analysis.

(2) DIAMOND: In our study, we initially conducted a homology-based search using BLASTn against the NCBI nt database and performed LCA annotation on the BLASTn output. However, this method resulted in a relatively low annotation rate, with only approximately 15% of the reads being identified. This low annotation rate is likely due to several factors,

including the fact that many taxa present in environmental samples do not have corresponding reference sequences in publicly available databases or that nucleotide sequences tend to be more variable than protein sequences. To address this limitation, we opted to use high-throughput DNA-to-protein alignment tool DIAMOND for our analyses. DIAMOND is significantly faster than BLAST and it can translate nucleotide sequences into protein sequences for alignment, which enhances its sensitivity and allows for the detection of more distant homologs (Buchfink, et al., 2021). This increased speed and sensitivity makes DIAMOND effective for metagenomic studies. Many studies have demonstrated the efficacy of DIAMOND in alignment and taxonomy classification within complex environmental samples (Simon, et al., 2019; Parks, et al., 2021). By integrating DIAMOND with LCA-based annotation tool BASTA, we were able to achieve a higher annotation rate and more comprehensive results, thereby improving the overall quality and reliability of our analysis.

- Buchfink, B., Reuter, K. and Drost, H.G., 2021. Sensitive protein alignments at tree-of-life scale using DIAMOND. *Nature methods*, 18(4), pp.366-368.
- Simon, H.Y., Siddle, K.J., Park, D.J. and Sabeti, P.C., 2019. Benchmarking metagenomics tools for taxonomic classification. *Cell*, 178(4), pp.779-794
- Parks, D.H., Rigato, F., Vera-Wolf, P., Krause, L., Hugenholtz, P., Tyson, G.W. and Wood, D.L., 2021. Evaluation of the microbial community profiler for taxonomic profiling of metagenomic datasets from the human gut microbiome. *Frontiers in Microbiology*, 12, p.643682.

Figure 1C, E and F: provide the number of samples for each category (Mid, Late, RA, Active, as well as the Soil types).

We have added the number of samples (Figs. 1C, 1E) and reads (Fig. 1F) for each category to Figure 1.

Figure 6: 17 categories in a stacked bar plot, many with very similar colours, might be a bit too much. Consider cutting some of the categories and provide the raw data in a supplementary table.

We have added data for all categories to a new supplementary table (Table S8) and redrawn Figure 6 using just seven categories (the most common taxa).

Comments from attachment 1:

“Adélie penguin mitochondrial nucleotide diversity recovered from the *sed*aDNA correlated with the number of breeding pairs in active colonies, allowing us to estimate the former size of abandoned colonies.” I assume you mean *sed*DNA as the correlation is based on active colonies? See my comments about this correlation based on seven (five) points).

Please see our earlier response in regard to the relationship between nucleotide diversity and colony size.

“Here, we report the first *sed*aDNA data generated from southern high-latitude terrestrial (non- aquatic) sediments.” Yes, if you narrow it down to this, I believe you are the first. However, I believe this race to be the first strip's the introduction for relevant background information as several relevant papers are not cited, e.g. Pienkowski et al. 2024 (<https://aslopubs.onlinelibrary.wiley.com/doi/full/10.1002/lol2.10395>), Ambrecht et al. 2022 (<https://www.nature.com/articles/s41467-022-33494-4>), Ficetola et al. 2018

(<https://www.science.org/doi/10.1126/sciadv.aar4292>), Weiss et al 2024

(<https://www.biorxiv.org/content/10.1101/2024.04.11.589015v1>)

We have removed mention of this being the “first *sed*aDNA data generated from...” from the introduction. We note that we already cite 10 papers in regard to “the utility of *sed*aDNA in understanding the long-term dynamics of aquatic (marine and lacustrine) ecosystems in Antarctica”, including the Armbrecht et al. 2022 reference provided by the reviewer. The Pieńowski et al. 2024 is a recent paper that is relevant here. We thank the reviewer for drawing our attention to it, and we include it as a citation now. We chose not to cite the Weiss et al. paper as it is still in preprint form, while Ficaretola et al. study is on the Kerguelen Islands, which at ~49 degrees South are well north of the Antarctic Circle and so are not particularly relevant in the context of our study.

Tables are not always numbered in order of appearance in the main text.

In the text, Table S3 was referred to before Table S2. We have corrected this.

“The percentage of reads from each sample that were assigned to Eukaryota generally declined both with increasing age since colony abandonment” Not shown in figure 1b.

In all plots within Figure 1, including 1B, the samples are coloured by age category (i.e. time since colony abandonment) in a consistent manner. This may not have been obvious, so we have now clarified it within the legend of Figure 1. The general decline in % Eukaryote reads is evident in Fig. 1B when looking at the distribution of samples from active colonies (white), where many samples have >0.02% eukaryote, through recently abandoned (green), late Holocene (yellow) to mid-Holocene (red), where no samples yielded >0.01%.

“As the majority of eukaryote reads identified were from taxa that live at the surface” Where is this shown?

Our apologies, this was shown in a previous figure that we removed prior to submission. We have added further clarification to the text here, stating that “80.6% of eukaryote reads resolved to phylum by the LCA approach were Chordata”.

“Fig. 1E” Capital or no capital? This varies.

We have now made these consistent.

“mosses (Bryopsida) “ Would be nice to add to figure 2

We have now added Bryopsida to Figure 2

“Within the stratigraphic context of individual pits the relative proportion of Adélie penguin mitochondrial DNA recovered was significantly greater from ornithogenic soils compared with minerogenic sediments” Thus, the results are highly impacted by spatial and temporal sampling and one needs to take this into account (see next comment). “It is conceivable that relative abundance of Adélie penguin DNA may correlate with local population density.” If the penguin DNA is spatially patchy distributed within the colony, one would need numerous systematic sampled sediments to correlate it to population density.

The point that we were trying to make here was that we know ornithogenic soils only form under high densities of Adélie penguin (i.e., within active colonies), whereas minerogenic soils reflect deposition outside the margins of a colony. Therefore, the fact that we find higher proportions of Adélie penguin mitochondrial DNA in the ornithogenic soils correlates with what we know about the relationship between local penguin population density and soil formation. This is essentially analogous to the assumption underpinning existing

geochemical proxies used for studying past population trends in Antarctic pygoscelid penguins. We have now clarified in the text that by local population density we are specifically talking about the number of birds within tens of metres of the excavation site.

“In support of this, we identified a positive relationship between the maximum *sedaDNA* mitochondrial *p* obtained from a sample within the uppermost 10 cm of sediment (i.e. the most recent past) and the current number of breeding pairs at active colonies (Fig. 3a; Fig. S2). ” I think this is the weakest part of the manuscript. No statistics are shown for this correlation, and there are only five known population sizes for seven samples.“ North Adélie Cove colonies were probably the smallest of the abandoned colonies in our study” What about Marble Point? “Stratigraphic patterns of mitochondrial *p* within individual sites provide indications of changes in colony size through time.“ I think this paragraph is ok as the wording is more careful than the previous paragraph.

Please see our earlier response in regard to the relationship between nucleotide diversity and colony size.

“Cnidarians represent a component of Adélie penguin diet that has historically been overlooked“ Very interesting and could be highlighted also in abstract.

We have now highlighted this finding in the abstract.

“Our study provides the first assessment of ancient DNA from Antarctic terrestrial (non-lacustrine) sediments” See earlier comment.

We have removed this claim from the abstract and introduction sections of the manuscript, but feel that it is a development that would be of interest to the wider readership and therefore worth retaining as part of the conclusion section. We have rewritten this slightly to focus on the implications of these new records in understanding the past: “Our study provides the first assessment of ancient DNA from Antarctic terrestrial (non-lacustrine) sediments and demonstrates the potential for using DNA from such samples to study spatiotemporal dynamics of both terrestrial and marine biodiversity at high southern latitudes.”

“Moreover, our results demonstrate that mitochondrial nucleotide diversity obtained from *sedaDNA* offers a new tool for reconstructing historic““

Please see our earlier response in regard to the relationship between nucleotide diversity and colony size.

“The sequencing data” What about tag look-up files? Without them the raw data are not informative.

We have now clarified that the deposited data do also include tag look-up files.

“Extended Data Fig. 1“ Can you increase letter size and add where samples for dates were taken?

Done

Reviewer 3

Thank you for the opportunity to review the manuscript "Spatiotemporal dynamics of Antarctic biodiversity revealed by terrestrial sedimentary ancient DNA".

I am not a specialist in DNA or sedaDNA, so my review is only concerning the radiocarbon dating and age modeling.

It is difficult to assess the degree of sediment reworking, which is also a concern raised by the authors. It would strengthen the dating results significantly, and possibly eliminate potential of sediment reworking, if the authors could provide more than one radiocarbon date from each layer. However, I do understand that this might not be practically possible if the sample material is limited.

We apologise for any confusion regarding the phrase “incorporation of younger eggshell into older sediment layers”. We did not intend to convey that this was due to sediment reworking or a pervasive issue with the stratigraphic integrity of the sites, but simply that some eggshell fragments from upper layers could have fallen down the sides of the pit while it was being excavated due to the unconsolidated nature of the sediments. We have now clarified this point in the manuscript. Accordingly, as radiocarbon dates were obtained on eggshell fragments that were sieved from bulk sediments collected during excavations, additional dates from each spit are unlikely to provide a resolution to this potential problem.

I suggest that the presentation of dating results should be made more clear, for example by annotating radiocarbon dating results on each stratigraphic profile in Extended Data Fig. 2. We have now annotated each stratigraphic profile in Extended Data Figs. 1 and 2 with the radiocarbon dating results.

As a general remark, I think the authors present a robust dataset and I appreciate that the authors take a conservative approach to dating results and use age groups rather than age-depth models.

Attachment:

This is an interesting approach to understanding long-term changes in species abundance/presence and biodiversity. SedaDNA has received little attention in at southern high latitudes. This study takes an impressive number of sediment samples in the Ross Sea region to determine temporal and spatial patterns in biodiversity.

I find the title to be a bit broad and I was expecting more biodiversity metrics/indices and comparisons given “biodiversity” was in the title. The study is mostly centered around Adélie penguins as the samples were collected at Adélie penguin colonies but some of the results are disjointed and read as a report rather than an interesting integration of detecting multiple species across the food web. I would be interested to see more results/statistics that compare/integrate the various species detections. I do find that the method seems promising, the results are interesting generally and specifically in understanding Adélie penguin diet/populations but sometimes the results are not totally interpretable to someone outside of this field.

I do want to note I am not able to comment on any of the sediment collection, radiocarbon dating, or DNA extraction methods as this is not within my expertise.

Specific comments

There were no page or line numbers making specific comments harder to point to.

In methods “(2) *Antarctic krill*:”. Crystal krill is also an important dietary item.

Figure 1: define “RA”. Describe “combined” in “c”? Are B/D totally necessary for the main text? Within the caption, you can you give time frames for Mid/Late/RA?

Can you describe what *Terminal (5') deamination rates* means?

Fig 1E. Interesting that mid is lower than late/RA. You say “*Mean fragment lengths differed significantly between all age groups*” but the stats don’t include a comparison between all groups.

No mention of collembola and eutardigrada in the text (Fig. 2).

Fig 2. “*Southern elephant seal was detected at all sites using the LCA approach, but these results could include other seal species*”. Can you explain a bit more? Is it that there is uncertainty around the presence or abundance of elephant seals detected, or something else?

Interesting about the cnidarian results. I see the mention of squid but what about octopus? These are occasionally seen in Adélie diets. Did you find any beaks from either species in the sediment samples?

The plankton figure/results seem unrelated to the main penguin story. Are some plankton ice-associated or relevant to fish vs. krill diet?